

# Research on 3D virtual vision matching based on interactive color segmentation

Yahui Wang[1], Haiwen Wang[1,2], Juan Jin[3] and Yingfeng Kuang[4]

[1] School of Humanities and Arts, Macau University of Science and Technology, Macau, China
[2] School of Art and Design, Wuhan Technology And Business University, Wuhan, Hubei, China
[3] School of Economics and Business Foreign Languages, Wuhan Technology And Business University, Wuhan, Hubei, China
[4] Xiangnan University, Chenzhou, Hunan, China

## ABSTRACT

Given the prevalent issues surrounding accuracy and efficiency in contemporary stereo-matching algorithms, this research introduces an innovative image segmentation-based approach. The proposed methodology integrates residual and Swim Transformer modules into the established 3D Unet framework, yielding the Res-Swim-UNet image segmentation model. The algorithm estimates the disparateness of segmented outputs by employing regression techniques, culminating in a comprehensive disparity map. Experimental findings underscore the superiority of the proposed algorithm across all evaluated metrics. Specifically, the proposed network demonstrates marked improvements, with IoU and mPA enhancements of 2.9% and 162%, respectively. Notably, the average matching error rate of the algorithm registers at 2.02%, underscoring its efficacy in achieving precise stereoscopic matching. Moreover, the model's enhanced generalization capability and robustness underscore its potential for widespread applicability.

Corresponding author
Yingfeng Kuang,
kk_kuang@outlook.com

## INTRODUCTION

Binocular stereo vision represents a prominent domain within 3D virtual vision, boasting extensive applications across various fields such as 3D measurement, autonomous vehicle navigation, 3D reconstruction, and target tracking (*Jiang, 2022*). The foundation of binocular stereo-vision technology encompasses crucial components such as camera calibration and 3D reconstruction, with the stereo-matching algorithm serving as the paramount stage within this comprehensive framework (*Li, Liu & Wang, 2022*). Consequently, the quest for precise and efficient stereo-matching algorithms enables the establishing of a robust binocular stereo-vision system and propels the advancement of 3D virtual vision.

Based on different optimization techniques, traditional stereo-matching algorithms can be categorized into global, local, and semi-global matching algorithms (SGM), with the latter being based on the former algorithm (*Yan, Yang & Zhao, 2022*). For example, in a study by *Kim, Kwon & Ko (2014)*, a proposed stereo-matching algorithm leverages confidence propagation to model the stereo-matching process as a Markov network.

Through iterative optimization *via* confidence propagation, the algorithm significantly enhances matching accuracy. *Bumsub et al. (2014)* introduces a stereo-matching algorithm based on steady-state matching probability, which dynamically adjusts the window size and achieves rapid matching speed, albeit with a trade-off of reduced performance in object edge regions. Additionally, *Chuang, Ting & Jaw (2018)* presents a progressive SGM cost aggregation scheme incorporating penalty adjustment and edge information, effectively preserving geometric edges and improving overall matching quality. However, traditional stereo-matching methods have inherent limitations and cannot incorporate carefully designed constraints, consequently impacting their performance to some extent. With the continuous development of deep learning, the fusion of convolutional neural networks and stereo-matching has yielded remarkable results (*Qi & Liu, 2022*). For instance, *Zbontar & LeCun (2016)* proposes a feature extraction approach using a weight-sharing network, simultaneous matching cost calculation, and similarity determination of extracted features from the left and right images *via* a fully connected layer. Another method presented in *Luo, Schwing & Urtasun (2016)* calculates feature similarity using dot product, replacing the earlier fully connected layer and enhancing operational efficiency. In recent years, these approaches have constantly improved from various perspectives, including algorithm stability, integration of semantic information, and operational efficiency enhancement (*Chen et al., 2015*). Nevertheless, these algorithms still require complex post-processing stages and manual design methods to handle exceptional areas and values, limiting their overall performance.

To minimize the need for manual intervention, a recent publication (*Mayer et al., 2016*) introduces an end-to-end stereo-matching model that incorporates the entire stereo-matching process within a unified network structure, eliminating the requirement for manual involvement. This model utilizes an "encoder–decoder" architecture and employs a large synthetic dataset to train the network model. Several algorithms that leverage large datasets have emerged to enhance algorithmic performance from various perspectives. For example, in *Kendall et al. (2017)*, 3DCNN makes more intuitive feature comparisons, combines contextual information, and integrates information from multiple angles to improve results. In *Zhang et al. (2021)*, a spatial pyramid pool is designed to expand the acceptance field of the network. Methods such as deep separable convolution, spatial pyramid pool and feature fusion are adopted to improve the performance, and the results are better than those of many advanced segmentation methods. In *Li, Zhao & Yan (2022)*, an attention mechanism was added to the stereo matching network, and combined with the stereo matching algorithm of hybrid extended convolution, the high matching accuracy and speed under unsupervised conditions were verified. In the *Du, El-Khamy & Lee (2019)*, it is proposed that hole convolution be used to extend the acceptance field of the network and an efficient feature extractor be adopted to propose a weight-stacked acyclic multiscale network, showing accurate parallax estimation. *Liang et al. (2021)* optimizes the residual to improve the parallax map and incorporates deeper feature information to enhance accuracy. In *Yao et al. (2021)*, images of different resolution levels are segregated and independently processed, with the resulting outputs subsequently fused. A novel adaptive matching approach is introduced in *Xu et al. (2022)*, introducing a new cost

function that can be broadly applied to most networks. The accuracy and convergence rate of the algorithm is improved in *Cheng et al. (2020)* through the introduction of a novel search method. *Liu, Yu & Long (2022)* integrates local features, edges, and semantic information to obtain more precise disparity maps.

Although significant strides have been achieved in stereo-matching algorithms, many of these methods are contingent upon pre-trained models utilizing extensive datasets. The necessity of pre-training on large datasets followed by fine-tuning presents an inherent complexity, as these models comprise numerous parameters and are not readily applicable to small-sample data. This study proposes an adaptive window stereo-matching algorithm based on an enhanced 3DUNet segmentation network to improve precision and efficiency in stereo matching. Experimental results reveal a notably low matching error rate. The key innovations of this study are as follows:

(1) Integrating a Residual module and Swim Transformer module into the 3DUNet depth neural network led to the development of the Res-Swim-UNet image segmentation model. This model facilitates accurate segmentation of stereo images.

(2) Introduction of the Soft Argmin operation to estimate the parallax of the Res-Swim-UNet segmentation results, thereby achieving precise stereo image matching.

(3) Experiments were carried out on different stereoscopic images to compare various models, and several different evaluation indicators achieved good experimental results, respectively.

The structure of this article unfolds as follows: 'Stereo Matching Algorithm based on Image Segmentation' outlines the proposed stereo-matching algorithm, while 'Experiment and Analysis' provides experimental validation of the algorithm's efficacy. Finally, 'Conclusion' offers a comprehensive summary of the article's content and presents prospects for future research directions.

## STEREO MATCHING ALGORITHM BASED ON IMAGE SEGMENTATION

### Image segmentation model based on improved 3DUNet

The 3DUNet is a renowned semantic segmentation network that serves as an extension of the UNet architecture. While its fundamental principle is akin to UNet, it distinguishes itself by utilizing 3D convolutions instead of 2D convolutions. This network has found widespread application in image segmentation across various domains (*Yu et al., 2022*).

Figure 1 illustrates the network structure, featuring convolution templates of sizes 64, 128, 256, 512, and 1,024. The feature map undergoes downsampling *via* $2 \times 2$ maximum pooling and applying the ReLU activation function. In the decoding network, the feature map is progressively upsampled and convolved to restore the original image size and channel count. The decoder and encoder are connected through a skip connection, enabling the fusion of feature maps. The fused feature map is further convolved and passed

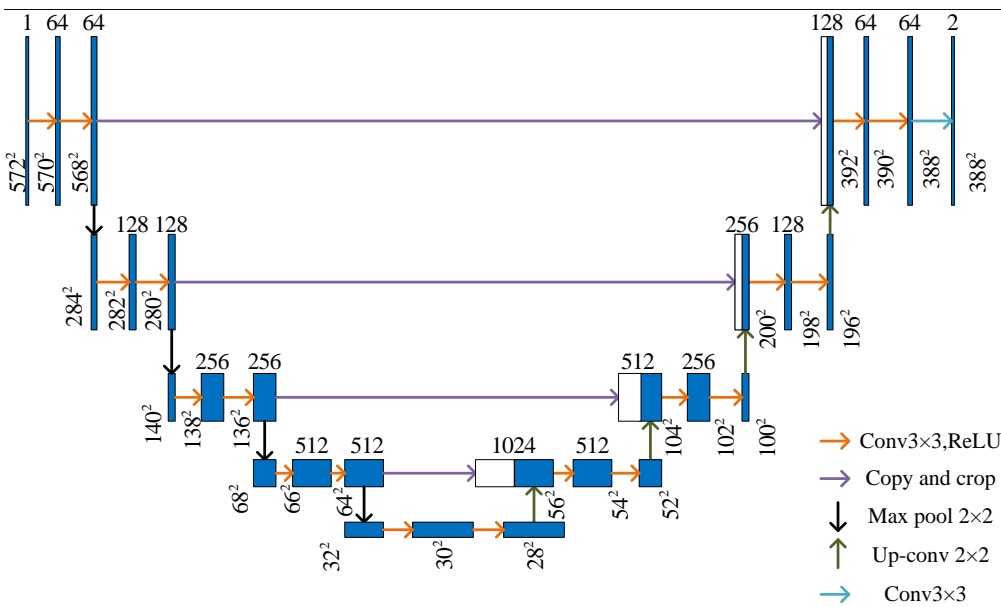

**Figure 1  3DUNet network structure.**

through a 1 × 1 convolution layer with C outputs, where C represents the number of channels or image segmentation categories.

The proposed Res-Swim-UNet model comprises an encoder, decoder, bottleneck layer, and skip connections. The encoder comprises six convolution layers, two residual modules, and four maximum pooling layers. This configuration facilitates extracting image features and downsampling, resulting in five feature maps. The skip connections concatenate the encoder's multiscale feature maps with the decoder. The decoder exhibits a symmetric structure to the encoder, comprising six convolution layers and two residual modules. The final layer incorporates a 1 × 1 × 1 convolutional layer and sigmoid activation functions to generate prediction probability maps. The bottleneck layer is situated at the lowest point of the U-shaped architecture, where the feature resolution is at its minimum. To optimize the computational cost of the Swim Transformer module, this study incorporates only two modules into the bottleneck layers, considering that the calculation cost of Swim Transformer modules increases linearly with resolution. This approach leads to an improved model with enhanced performance. Figure 2 provides a detailed illustration of the model's structure.

The conventional Transformer relies on multi-head self-attention modules to establish global interdependencies, enabling the system to extract and analyze global information better. By contrast, the Swim Transformer is designed on the principle of a mobile window and is composed of contiguous sub-modules. Figure 3 shows a detailed illustration of its structure.

Each submodule within the Swim Transformer comprises a normalization layer, a multi-head self-attention module, a residual connection, and a two-layer Multilayer Perceptron (MLP). Window Multi-head Self-Attention (W-MSA) and Shifted Window

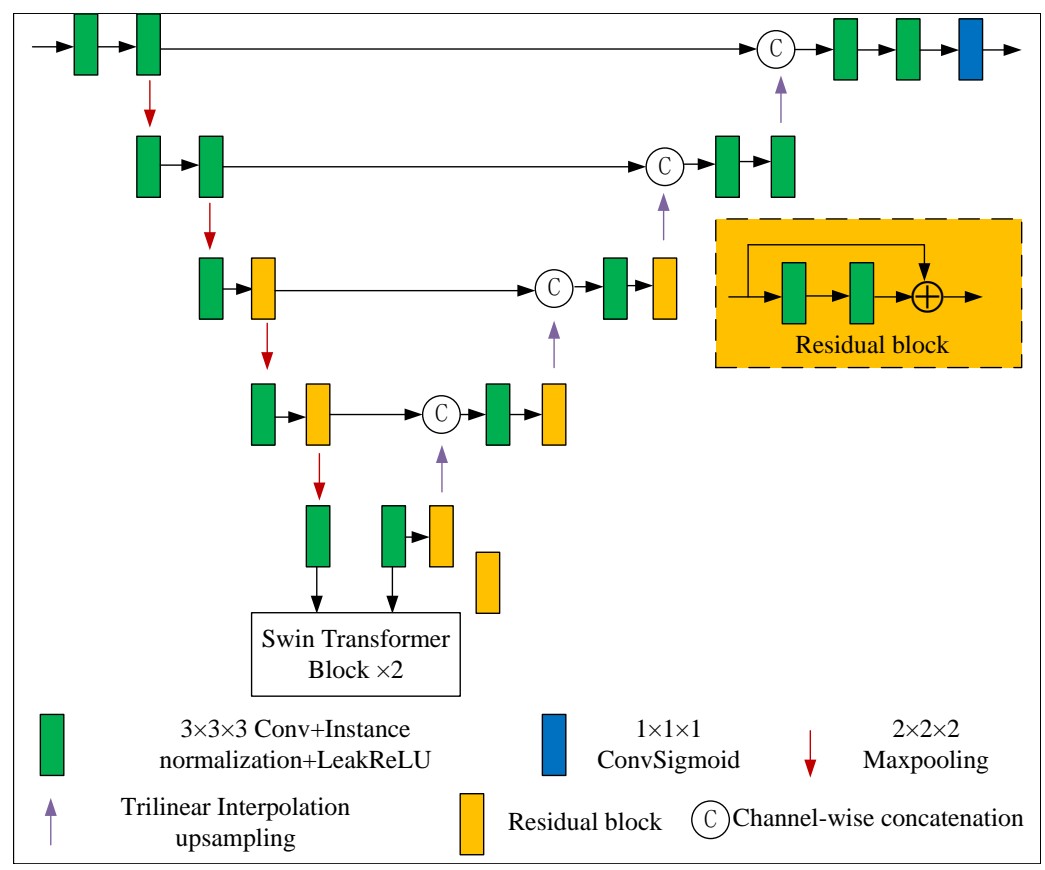

**Figure 2  Res-Swim-UNet model structure.**

Multi-head Self-Attention (SW-MSA) are employed for two consecutive submodules of the Swim Transformer. Formulas (1) to (4) calculated the Swim Transformer module in detail.

$$\hat{z}^l = WMSA\left[LN\left(z^{l-1}\right)\right] + z^{l-1} \tag{1}$$

$$z^l = MLP\left[LN\left(\hat{z}^l\right)\right] + \hat{z}^l \tag{2}$$

$$\hat{z}^{l+1} = SWMSA\left[LN\left(z^l\right)\right] + z^l \tag{3}$$

$$z^{l+1} = MLP\left[LN\left(\hat{z}^{l+1}\right)\right] + \hat{z}^{l+1} \tag{4}$$

where, $\hat{z}^l$ respectively, represent the W-MSA module and MLP module for the module output, LN represents layer normalization, where the self-attention module can be

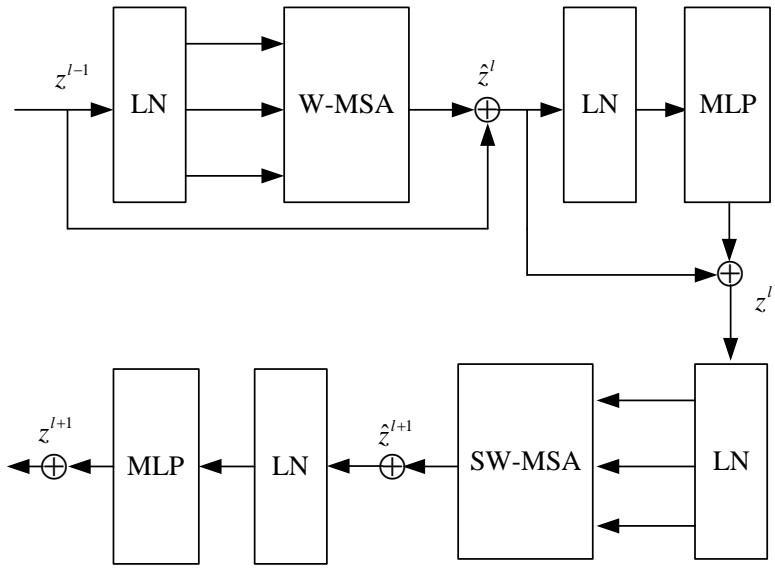

**Figure 3** The module structure of the Swim Transformer.

expressed as Formula (5).

$$\text{Attention}(Q, K, V) = \text{Softmax}\left(\frac{QK^T}{\sqrt{d}} + B\right)V \tag{5}$$

where, $Q, K, V \in R^{M^2 \times d}$ represents three matrices, which are obtained from the input characteristic map through three convolution layers. $K^T$ is the transpose matrix of $K, d$ is the scaling factor set to 64 in the experiment, $B$ is a learnable offset parameter.

## Parallax regression

The Res-Swim-UNet model utilizes regression to estimate the segmentation results and employs the fully differentiable Soft Argmin operation (*Li, Zhao & Yan, 2022*) to yield smooth disparity estimation outcomes. This operation involves calculating the probability of each pixel in the segmented image belonging to various parallax values. The matching cost volume is then processed using 3D CNN and upsampled to obtain the matching cost for each pixel across all parallax values. The prediction cost is negated and regularized using a Softmax operation, resulting in the probability of each pixel belonging to different parallax values. Finally, the parallax values are weighted and summed using the corresponding probability values to determine the parallax value at each pixel, as depicted in Formula (6).

$$d = \sum_{d=0}^{D\text{max}} d \times \sigma(-Cd) \tag{6}$$

where $d$ represents the predicted parallax value, $Cd$ represents Matching costs under parallax $d, \sigma(.)$ represents the Softmax operation, and its mathematical expression is

shown in Formula (7).

$$\sigma(zj) = \frac{e^{zj}}{\sum_{k=1}^{K} e^{zk}}$$ (7)

where $j = 1, 2, \ldots, K$. The result of the converted output by Softmax is always in the range [0, 1], and the sum of all results is equal to 1, so they show a probability distribution.

## Loss function

Given that the smooth L1 loss function demonstrates strong robustness and low sensitivity to outliers (*Xu et al., 2022*), this study has adopted it as the fundamental loss function, as presented in Formula (8).

$$L(d.\hat{d}) = \frac{1}{N} \sum_{n=1}^{.N} L_1^s(dn, \hat{d}n)$$ (8)

$$L_1^s(x) = \begin{cases} 0.5x^2, |x| < 1 \\ |x| - 0.5 \end{cases}$$ (9)

where $N$ represents the total number of pixels, $dn$ represents the true parallax value, $\hat{d}n$ and represents the predicted parallax value. In general, it minimizes the sum $L(d.\hat{d})$ of the absolute differences between the target value $dn$ and the estimate $\hat{d}n$.

A deep supervision training approach is deployed to supervise the network's final output and the results obtained from intermediate levels of the network. More precisely, apply a corresponding loss function, such as a cross-entropy loss function, to the output of each layer, compare the loss value of each layer to the real label, perform parallax regression on the volume output of each encoding and decoding structure, and calculate the loss value, backpropagate according to the loss value, update the network parameters, and thus improve the performance of the entire network. The final loss value is the weighted sum of the loss values from each level, as illustrated in Formula (10).

$$Loss = \sum_{i=1}^{M} wiLi(d.\hat{d})$$ (10)

were $wi$ represents the weight of losses at different levels, $M$ represents the number of supervised levels. After experiments, $M = 3$ each layer's corresponding weight parameters are between 0.5–1.0.

## EXPERIMENT AND ANALYSIS

### Data set and parameter setting

To evaluate the effectiveness and robustness of the proposed algorithm, Comparative experiments were carried out on different stereoscopic image pairs (https://zenodo.org/records/45114). All the comparison experiments were conducted under the same environment and hyperparameter settings.

**Table 1  Matching parameters.**

| Parameter | $\alpha$ | $\beta$ | $\gamma$ | $\lambda c$ | $T_{Gx/y}$ | $T_{Census}$ |
|---|---|---|---|---|---|---|
| Value | 0.10 | 0.45 | 0.45 | 9.6 | 8 | 8 |

**Table 2  Comparison of model segmentation performance.**

| Network name | IoU | mPA | FPS |
|---|---|---|---|
| UNet | 0.886 | 0.929 | 33.3 |
| UNet++ | 0.901 | 0.942 | 35.2 |
| 3DUNet | 0.923 | 0.954 | 49.1 |
| ResUNet | 0.936 | 0.972 | 52.3 |
| Ours | 0.965 | 0.988 | 65.6 |

All experiments were conducted using the Python language, and computation and the OpenCV visual library facilitated compilation. All pixel values were normalized to the range [0, 1.0]. Parameters utilized in the experiments are listed in Table 1. To calculate the mismatching rate, detailed data from mismatching pixels was obtained by comparing each pixel in the resulting parallax map to the corresponding pixel in the standard parallax map, which is calculated by Formula (11):

$$PPBM = \frac{1}{N} \sum_{(x,y)} \left[ \left| dc(x,y) - dt(x,y) \right| > \delta d \right]. \tag{11}$$

The parameters in the equation have the same meanings as those in Table 1, where $N$ is the number of effective pixels in the image area, $dc(x,y)$ is the disparity map calculated for stereo matching algorithm; $dt(x,y)$ is the true parallax map provided for the dataset, $\delta d$ is the parallax threshold, which is taken as 1 in the experiment, that is, when the difference between the parallax value calculated by the stereo matching algorithm and the true parallax value is greater than 1, the pixel point is regarded as a mismatched point. The specific step is to give two graphs with different marks, search for the most matched points on the polar line, calculate the similarity of each search window, and finally calculate the parallax.

## Comparison of segmentation performance

The segmentation model's performance plays a vital role in the overall matching effect of the proposed algorithm, given its sequential segmentation and matching approach. To validate the effectiveness of the introduced residual module and Swim Transformer module, a comparison was conducted between the Res-Swim-UNet model proposed in this study and other models, including UNet, UNet++, 3DUNet, and ResUNet. The evaluation of segmentation performance encompassed metrics such as the intersection-over-union ratio (IoU), mean pixel accuracy (mPA), and frames per second (FPS) (*Yu et al., 2022*; *Rahman et al., 2022*). The comprehensive comparison results of the five models are presented in Table 2, while Fig. 4 depicts the mPA curve over 600 iterations.

Table 2 and Fig. 5 illustrate that the model proposed in this study outperformed the other models across all three evaluation metrics. Although the improvement in mPA was

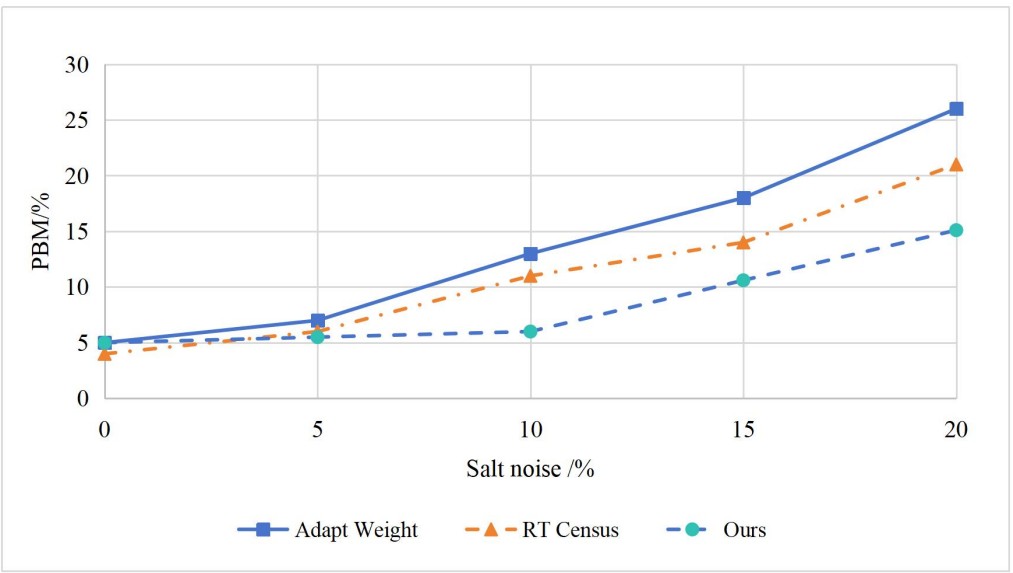

**Figure 4** Comparison of different salt and pepper noise.

not particularly significant compared to the other models, the model in this article exhibited superior generalization ability and robustness. By incorporating the residual connection and Swim Transformer, the network in this article demonstrated improvements of 2.9% in IoU and 162% in mPA. Moreover, it achieved an FPS of 65.6/s, representing a 25.4% enhancement over the ResUNet network, which solely utilized the residual module for optimization. These results conclusively demonstrate the effectiveness of the Swim Transformer module proposed in this article, as it successfully enhances the accuracy and efficiency of the image segmentation model.

## Comparison of matching performance of different algorithms

To assess the matching performance of the algorithm proposed in this article, a comparison was conducted with the classic Adaptive Weight and RT Census algorithms (*Deng et al., 2023*; *Chen et al., 2023*). Table 3 presents the mismatch rates of the various algorithms. Based on the test results, the proposed algorithm achieved an average mismatch rate of 2.04%, surpassing the performance of the other two algorithms. Thus, the proposed model effectively enhances the matching accuracy.

In order to further examine the higher-resolution image matching, we perform stereo matching on standard stereo image pairs of Adirondack (resolution: 718 × 496), Pipes (resolution: 735 × 485), Motorcycle (resolution: 741 × 497), and ArtL (resolution 694 × 554), employing the error matching rate of non-occluded area (Nonocc), all areas (All), and discontinuous area (Disc) as evaluation indices. The experimental results are tabulated in Table 4. Our observations from Table 4 lead to the conclusion that the matching accuracy of the algorithm in this article is high and that it is also suitable for stereo matching of high-resolution images.

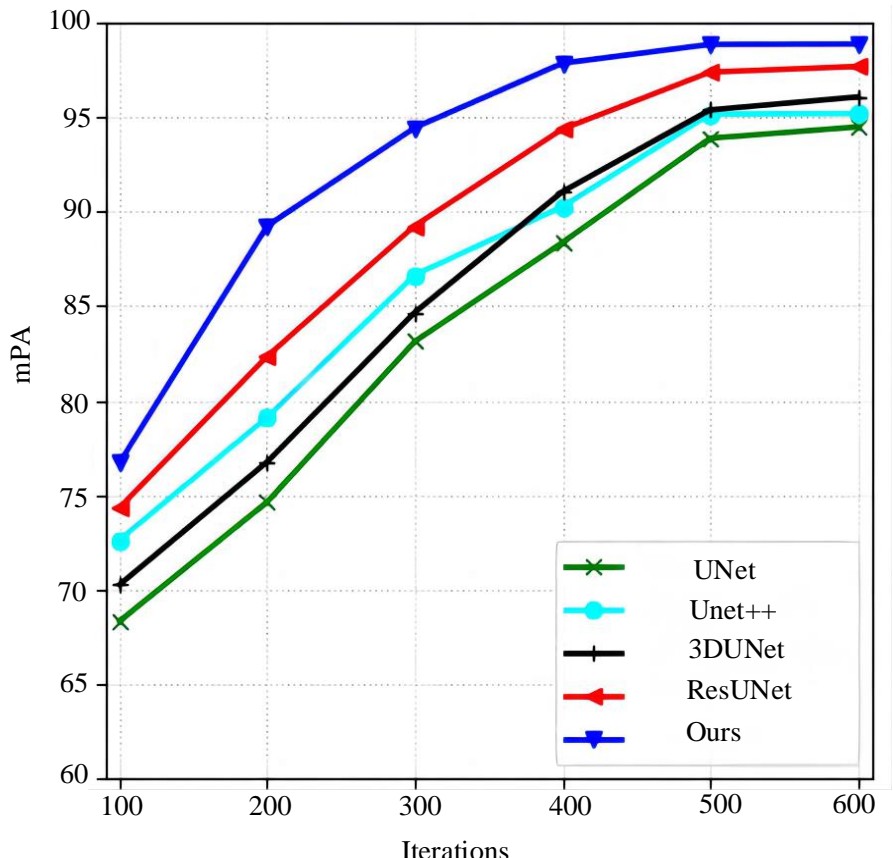

**Figure 5** MPA curves of different models.

**Table 3** Comparison of mismatch rate.

| Algorithm name | Conse | Teddy | Venus | Tsukuba | Average |
|---|---|---|---|---|---|
| Adapt Weight | 16.42 | 7.21 | 7.86 | 8.32 | 9.95 |
| RT Census | 10.12 | 12.46 | 0.89 | 2.85 | 6.58 |
| Ours | 3.99 | 3.21 | 0.67 | 0.23 | 2.02 |

**Table 4** Matching results of adirondack, pipes, motorcycle, artL.

| Evaluating indicator | Adirondack | Pipes | Motorcycle | Artl |
|---|---|---|---|---|
| Nonocc | 0.0109577 | 0.0492873 | 0.224144 | 0.0281561 |
| Disc | 0.0844523 | 0.467321 | 0.207226 | 0.377693 |
| All | 0.0251666 | 0.1571 | 0.0618214 | 0.14966 |

## Anti-noise test

In addition, 5%, 10%, 15%, and 20% pepper and salt noise and Gaussian noise with standard deviation were added to the standard test images of Tsukuba, Venus, Teddy, and Cones. The two algorithms above, compared in 'Comparison of matching performance of

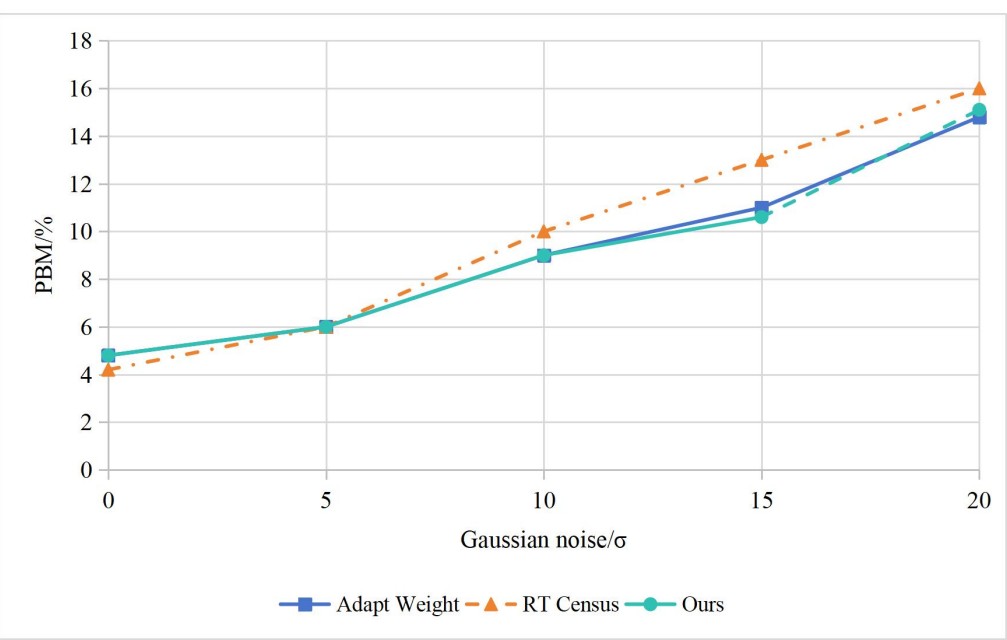

**Figure 6  Comparison of different standard deviation Gaussian noise.**

different algorithms', were utilized to calculate and obtain the unoptimized disparity map and subsequently compute the mismatch rate. The resulting experimental outcomes are presented in Figs. 5 and 6.

Based on the data presented in Figs. 5 and 6, it can be observed that the Adapt Weight algorithm consistently exhibited a high mismatch rate when subjected to salt and pepper noise, indicating its vulnerability to noise interference. On the other hand, the proposed algorithm showcased superior robustness to such noise. In Gaussian noise, both the proposed algorithm and the RT Census algorithm outperformed the Adapt Weight algorithm. In conclusion, the algorithm proposed in this article demonstrates enhanced noise tolerance without compromising the accuracy of the resulting disparity map, surpassing the performance of both the Adapt Weight and RT Census algorithms.

## Discussion

The advancement of convolutional neural network architectures in image segmentation, coupled with the availability of sizeable standardized stereo datasets, has led to extensive research on stereo-matching algorithms based on deep learning. Many of these algorithms now achieve accuracy comparable to that of depth sensors. Compared to traditional stereo matching methods that rely on manually designed features, deep learning-based 3D matching algorithms have significantly improved accuracy. Considering the parallax continuity constraint in stereo matching, which states that there should be no abrupt changes in parallax between the center pixel and its neighboring pixels, except at the image boundary, the image can be divided into segments assuming continuous parallax within each segment. The encoder–decoder network based on 3D convolution offers advantages

such as a simple structure, strong scalability, and ease in balancing precision and speed. Therefore, this study combines the classical image segmentation network, 3DUNet, with the stereo-matching algorithm. A new parallax computation network is proposed by incorporating residual and Swim Transformer modules to adjust the network structure and optimize performance. This fusion approach has two advantages in experiments: it ensures adherence to the parallax continuity constraint. It reduces the algorithm's time complexity by operating at the superpixel level rather than the pixel level. The encoder–decoder network, also known as the hourglass structure, exhibits powerful feature extraction and data dimension reduction capabilities, making it well-suited for stereo-matching tasks. With the continuous progress of social science and technology, three-dimensional matching technology is advancing rapidly. The improvement in the accuracy and speed of matching algorithms has expanded their application scenarios. Against this backdrop, studying the variations in stereo matching is significant. Stereo matching is crucial in obtaining depth information through image matching in 3D reconstruction, stereo navigation, non-contact ranging, and various other technologies. Although stereo matching is widely utilized, numerous unresolved issues remain, making it a challenging and prominent topic in computer vision in recent years.

Although the experimental comparison was not carried out on different data sets in this article, the model could learn more data patterns when there were enough data samples, select meaningful features in feature selection, and reduce overfitting. The pre-trained model was adopted as initialization, and fine-tuning was performed from convergent points to increase the model's generalization ability.

As an engineering problem, stereo matching involves various factors that affect its accuracy and speed during implementation. No single complex algorithm can address all aspects of stereo matching. This article focuses on the core steps of image pixel segmentation and matching in stereo matching and improves the image segmentation network by integrating it into the stereo matching algorithm. Experimental results have demonstrated the effectiveness of the proposed approach. Therefore, it can be concluded that the algorithm presented in this article can assist researchers in establishing more effective binocular stereo-vision systems and contribute to advancing related fields.

## CONCLUSION

The article introduces a novel adaptive window binocular stereo-matching algorithm incorporating image segmentation. The algorithm leverages the Residual and Swim Transformer modules within a 3DUN segmentation model to accurately segment the input images. It further utilizes the Soft Argmin operation to estimate the parallax of the segmented images, leading to improved stereo-matching results. Compared with the traditional plane matching algorithm, the three-dimensional matching method in this article plays a crucial role in obtaining depth information through image matching. The experimental results show that compared with other models, the proposed algorithm has better results on different evaluation indexes, improving the image segmentation network and having higher segmentation accuracy and lower mismatch rate, which ensures

accurate and stable stereo-matching performance. Experimental results demonstrate that the proposed algorithm achieves high segmentation accuracy and low mismatch rates, guaranteeing precise and stable stereo-matching performance. In future work, the authors aim to enhance the segmentation model to achieve even more accurate results. Additionally, they seek to improve the quality of stereo matching further. These advancements will contribute to developing more sophisticated and effective binocular stereo-vision systems.

## ACKNOWLEDGEMENTS

The author would like to thank the anonymous reviewers for their valuable comments on this article.

### Funding

The authors received no funding for this work.

### Competing Interests

The authors declare there are no competing interests.

### Author Contributions

- Yahui Wang conceived and designed the experiments, analyzed the data, prepared figures and/or tables, and approved the final draft.
- Haiwen Wang conceived and designed the experiments, performed the computation work, authored or reviewed drafts of the article, and approved the final draft.
- Juan Jin performed the experiments, performed the computation work, prepared figures and/or tables, and approved the final draft.
- Yingfeng Kuang performed the experiments, analyzed the data, authored or reviewed drafts of the article, and approved the final draft.

### Data Availability

3D Matching of resource vision tracking trajectories are available at Zenodo: Konstantinou, E., & Brilakis, I. (2016). 3D Matching of resource vision tracking trajectories [Data set]. Zenodo. https://doi.org/10.5281/zenodo.45114.

### Supplemental Information

Supplemental information for this article can be found online at http://dx.doi.org/10.7717/peerj-cs.2114#supplemental-information.

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
