# Peer review of "Research on 3D virtual vision matching based on interactive color segmentation"

_PeerJ Computer Science, doi:10.7717/peerj-cs.2114_

## Round 0.1 · original submission · Major Revisions

· Academic Editor

Major Revisions

Reviewers emphasize the need for improved clarity, methodology, and presentation in the stereo-matching algorithm manuscript. They suggest enhancing the presentation of the work's contribution, conducting comprehensive comparisons with existing methods, and refining explanations of equations and algorithms.

Also, language and formatting improvements are needed, clarity in evaluation methodology, discussion on training losses, and consideration of model complexity are needed.

Please see both reviewers' detailed comments.

**Language Note:** The review process has identified that the English language must be improved. PeerJ can provide language editing services - please contact us at [email protected] for pricing (be sure to provide your manuscript number and title). Alternatively, you should make your own arrangements to improve the language quality and provide details in your response letter. – PeerJ Staff

Reviewer 1 ·

Basic reporting

The manuscript requires modifications in language expression, particularly in replacing Chinese characters in formulas and eliminating redundant spaces in references. Authors should ensure accuracy and consistency in language usage and formatting throughout the paper to enhance readability and professionalism.

Experimental design

1. The manuscript lacks detailed elucidation regarding the criteria for identifying pixels as mis-matched points. Given the reliance on evaluation metrics such as IoU, mPA, and FPS, it's imperative to establish a clear connection between these metrics and the calculation of matched points. A thorough explanation in this regard would provide readers with a more nuanced understanding of the evaluation methodology employed.

2. While the manuscript adopts a deep supervised training approach, there remains a notable absence of comprehensive discussion on the calculation of training losses at different levels. Further exploration into the intricacies of calculating losses across various levels during model training is warranted. Such an elucidation would shed light on the underlying mechanisms driving model optimization.

3. The introduction of both Residual and Transformer modules adds a layer of complexity to the model architecture, potentially resulting in increased parameters and computational costs. This heightened complexity could pose resource challenges during both training and inference phases.

Validity of the findings

Certain statements in the article, such as "When the distance between the signal transmitting device and the signal receiving device is too long, it is difficult to find the distance between the two devices only through the single signal," are confusing and lack relevance to the model under discussion. Authors should review and revise such statements to ensure clarity and relevance to the study's focus.

Additional comments

1. Despite demonstrating promising performance on specific datasets, the manuscript fails to adequately address the critical issue of model generalization across diverse datasets. This lack of discussion raises concerns regarding the model's broader applicability and portability in real-world scenarios. Therefore, it is imperative for the authors to delve into a comprehensive analysis of the model's generalization performance. Furthermore, the manuscript should offer insights into potential strategies or enhancements to improve generalization capabilities, thereby ensuring the model's efficacy across varied datasets and settings.

2. While the manuscript addresses the importance of HCI plane visual image color control, the summary of relevant work in this field is insufficient. Authors should provide a more comprehensive overview of existing research to contextualize their study and highlight the gaps or limitations that their work aims to address.

3. The author should emphasize the contribution of this research in the abstract, introduction, and conclusion sections. Clearly articulating the unique contributions and significance of the study will help readers understand its value and relevance within the broader research domain.

Cite this review as

Reviewer 2 ·

Basic reporting

The manuscript presents a novel image segmentation-based stereo-matching algorithm aimed at addressing the challenges of inadequate precision and efficiency observed in current algorithms. While the proposed methodology is well-described, there is room for improvement in emphasizing the novelty of the addressed problem. Authors should provide additional context or examples to highlight the uniqueness of the problem addressed, thereby enhancing the significance and originality of their work.

Experimental design

While the experimental results demonstrate a commendable average matching error rate of 2.02%, the paper lacks a thorough comparison with other state-of-the-art methods. Authors should conduct a more comprehensive comparison with existing approaches to better elucidate the strengths and limitations of their proposed algorithm, providing readers with a clearer understanding of its competitive advantages.

Validity of the findings

The background section requires revision to provide a more coherent and reader-friendly narrative. Instead of presenting a mere list of references, authors should contextualize the relevant literature, offering insights into how prior works inform the current study. This will enhance the readability and comprehensibility of the manuscript for readers unfamiliar with the field.

Additional comments

The explanations of equations and algorithms need to be improved for better clarity and understanding. Authors should ensure that each equation and algorithm is thoroughly described, providing readers with a clear understanding of their purpose, methodology, and significance within the context of the study.

 Overall, while the study presents a promising approach to stereo matching, addressing these concerns would significantly strengthen the manuscript and its contribution to the field. By emphasizing the novelty of the addressed problem, conducting a thorough comparison with existing methods, revising the background section, and improving the clarity of equations and algorithms, authors can enhance the impact and readability of their work.

 It is crucial to conduct a comprehensive comparison with existing methods to elucidate the strengths and limitations of the proposed algorithm fully. Authors should provide a detailed comparative analysis to help readers better understand the competitive advantages of their approach.

 Enhancing the clarity of explanations for equations and algorithms is essential for improving reader comprehension. Authors should strive to provide clear and concise descriptions, ensuring that readers can grasp the purpose and significance of each mathematical concept or computational procedure presented in the manuscript.

Overall, the paper presents a promising contribution to the field, but revisions are needed to address clarity, methodology, and presentation issues. A major revision focusing on providing detailed explanations of the technical aspects, discussing practical implications, and exploring future research directions is recommended.

Cite this review as

---

## Round 0.2 · accepted · Accept

· Academic Editor

Accept

Both reviewers have confirmed that their comments are addressed.

Reviewer 1 ·

Basic reporting

Article is now well shaped.

Experimental design

Article is now well shaped

Validity of the findings

Article is now well shaped

Additional comments

Article is now well shaped

Cite this review as

Reviewer 2 ·

Basic reporting

no comment

Experimental design

no comment

Validity of the findings

no comment

Additional comments

no comment

Cite this review as